# Solid matrix priming improves cauliflower and broccoli seed germination and early growth under the suboptimal temperature conditions

Wu Ling-Yun[1,2]*, Yan Jun[1,2], Huang Zhi-wu[1,2], Wan Yan-Hui[1,2], Zhu Wei-Min[1,2]*

**1** Horticultural Research Institute, Shanghai Academy of Agricultural Sciences, Shanghai, China, **2** Shanghai Key Lab of Protected Horticultural Technology, Shanghai, China

* wulingyun@saas.sh.cn (WL); wmzhu69@126.com (ZW)

**Data Availability Statement:** All relevant data are within the paper.

**Funding:** Weimin Zhu and Lingyun Wu received the Shanghai Agriculture Applied Technology

## Abstract

Seed priming is an effective method for imparting stress tolerance to plants. This study aimed to analyze the effects of solid matrix priming (SMP) on cauliflower and broccoli seed germination and early seedling growth under suboptimal temperature conditions. The SMP method used in this study included the following steps: (1) mixing seeds with vermiculite and water at a ratio of 2:3:2.5 (*w*/*w*/*v*) and incubating for 2 days in the dark at 20°C; (2) drying the SM-primed seed; (3) germinating the SM-primed and the nonprimed seeds at 10, 15, 20, and 25°C; (4) analyzing the antioxidant enzyme activities of SM-primed and nonprimed germinating broccoli and cauliflower seeds in the early germination stage at 10, 15, 20, and 25°C; and (5) testing the emergence of SM-primed and nonprimed control seeds in the early spring glasshouse. The results showed that the SMP improved seed germination vigor and early seedling growth and increased the activities of peroxidase and ascorbate peroxidase in the germinating cauliflower and broccoli seeds under the suboptimal temperature conditions in the early germination stage compared with nonprimed seeds. It was observed that the suboptimal temperature conditions (i.e., 10 and 15°C) suppressed SM-primed and nonprimed seed germination and early seedling growth of cauliflower and broccoli. Inside a greenhouse, the SMP improved the emergence of cauliflower and broccoli seeds during the early spring season. SMP is an effective method for improving seed germination and the emergence of cauliflower and broccoli under suboptimal temperature conditions.

## Introduction

Cauliflower (*B. oleracea* L. var. *botrytis*) and broccoli (*B. oleracea* var. *italica*) are important vegetable species widely cultivated and consumed. Temperature is an important factor that can affect seed germination and emergence significantly. Cauliflower and broccoli seeds are often sown in greenhouse, seed beds, or open field in winter or in early spring. Therefore, their germination and seedling emergence are one of the major concerns of farmers. Seed priming is a method that can be used to improve seed vigor by enhancing seed germination rate (GR) and uniformity of emergence [1, 2]. Seed priming is a physiological method that involves seed

Development Program, China(Grant No. T20190112 and Grant No.G2016060105). The funders had no role in study design,data collection and analysis, decision to publish, or preparation of the manuscript.

**Competing interests:** The authors have declared that no competing interests exist.

**Abbreviations:** APX, Ascorbate peroxidase; BA, benzyladenine; CAT, catalase; EDTA, ethylene diamine tetraacetic acid; EI, emergence index; VI, vigor index; GR, germination rate; GV, germination vigor; MET, mean emergence time; MGT, mean germination time; NP, nonprimed; PEG, polyethylene glycol; POD, peroxidase; ROS, reactive oxygen species; SA, salicylic acid; SMP, solid matrix priming; SOD, Superoxide dismutase.

hydration and is effective enough for enhancing seed germination, early seedling growth, and yield under stressed and nonstressed conditions but is insufficient to allow radical protrusion [3]. Seed priming activates pre-germinative metabolic events to hasten rapid and uniform seed germination. Several studies have indicated that seed priming can induce enzyme activities and increase the production of antioxidants and biosynthesis of carbohydrates, phytohormones, late embryogenesis abundant proteins, and many other proteins, including heat shock proteins [3–6]. In addition, priming can enhance DNA replication and repair as well as respiratory associated pathways [7]. Seed priming has also been shown to reduce cellular damages during water imbibition [3]. Consequently, the seedlings derived from primed seeds often exhibit enhanced tolerance to abiotic stresses [8].

Seed priming is an effective method to improve plant tolerance to low temperatures. For example, priming seeds with $CaCl_2$ can improve wheat and maize chilling tolerance in late-sown conditions [9]. Seed priming can significantly improve bitter gourd seed germination vigor (GV) and upregulate the levels of multiple anti-oxidative enzymes [10]. Primed tobacco seeds with putrescine significantly increased seed GV and seedling biomass compared with nonprimed control during chilling stresses [11]. Osmopriming has been shown to improve spinach germination at 5 and 20°C [12]. Demir and Mavi reported that priming watermelon seeds with $KNO_3$ increased seed emergence and improved seedling growth under greenhouse conditions in early spring [13]. Seed priming with the incorporation of SA into the $KNO_3$ solution improved the low-temperature performance of eggplant seeds and subsequent seedling growth [14]. Sorghum seed priming with the inclusion of BA into PEG solution showed an improvement in germination attributes under low-temperature conditions [15]. Solid matrix priming (SMP) combined with *Trichoderma viride* has been shown to improve okra seed emergence and fruit productivity under low-temperature conditions [16].

This study was performed to evaluate the effects of SMP on cauliflower and broccoli seed germination and early seedling growth under suboptimal temperature conditions, and to investigate the physiological and biochemical changes in the germinating seeds in the early germination stage.

## Materials and methods

### Seed material and experimental treatments

The seeds of cauliflower cv. Husong 85 and broccoli cv. No. 5 Hulv were obtained from the Shanghai Academy of Agricultural Sciences, Shanghai, China. For SMP, these seeds were mixed with vermiculite and water at a ratio of 2:3:2.5 (*w/w/v*) and then incubated for 2 days in the dark at 20°C. The primed seeds were dried at 25°C for 24 h and then at 30°C for 24 h prior to further use. Seeds without SMP treatment were used as controls, and the initial moisture content was 4.2%.

### Seed germination test

SM-primed or nonprimed control seeds were placed on two layers of filter paper pre-wetted with 7 mL of $H_2O$ inside plastic boxes ($12 \times 12 \times 6$ cm$^3$). The plastic box was placed inside a germination chamber (Zhejiang Top Instrument Co., Ltd.) set at 10, 15, 20, and 25°C for 7 days. Each treatment involved 50 seeds, and the experiment was conducted 3 times using a complete randomized design. The germinated seeds were counted daily. On the seventh day after incubation, the fresh weight and root length of 10 germinated seedlings were measured individually to represent each treatment. The seed GV, GR, vigor index (VI), and mean germination time (MGT) were calculated as per standard procedures [17].

### Analysis of antioxidant enzyme activities

SM-primed and nonprimed seeds were germinated for 17 h for broccoli and 24 h for cauliflower at 10, 15, 20, and 25˚C, harvested, and then analyzed for antioxidant enzyme activities. The harvested SM-primed or nonprimed germinating seeds were ground and extracted in ice-chilled 50mM potassium phosphate buffer, pH 7.0, with 0.1mM ethylene diamine tetraacetic acid (EDTA). The extracts were centrifuged at 17,000$g$ for 20 min at 4˚C. The supernatant of each sample was collected to determine the enzyme activities.

The superoxide dismutase (SOD) activity was determined as described [18]. The reactions were conducted in individual tubes and illuminated for 25 min at 25˚C. The absorbance was recorded at 560 nm. The reactions in the tubes without illumination were used as controls. The catalase (CAT) activity was measured as described previously by Cakmak and Marschner [19]. The decrease in absorbance at 240 nm for $H_2O_2$ was measured for each reaction and used to represent the CAT activity. The peroxidase (POD) activity was determined by measuring the oxidation of guaiacol in the presence of $H_2O_2$ and the increase in absorbance at 470 nm over 1-min intervals. The ascorbate peroxidase (APX) activity was determined as previously described [20]. The oxidation of ascorbate was induced by $H_2O_2$, and the decrease in absorbance at 290 nm every 2 min was monitored.

Each treatment had three biological replicates in this study.

### Emergence test

The emergence test using SM-primed and nonprimed control seeds was performed in a glasshouse. For each treatment, 2 biological replicates with 50 seeds each were tested in the glasshouse. In this study, the daily minimum air temperature ranged from 3.9 to 11.8˚C, and the daily maximum air temperature ranged from 11.4 to 34.7˚C. The seedling emergence was recorded daily for 16 days, and a seed was considered as emerged when its hypocotyl appeared above the ground surface. The mean emergence time (MET) and emergence index (EI) were calculated as described below.

$EI = \sum (Et/Dt)$, where $Et$ is the number of emerged seeds in $t$ days and $Dt$ is the number of the corresponding emergence days.

$$MET = \sum(Et \times Dt)/\sum Et.$$

### Statistical analysis

The data were analyzed for the statistical differences using SPSS v.16. Duncan's multiple range test was used for one-way analysis of variance to determine the difference among different treatments.

## Results and discussion

### Effects on seed germination and early seedling growth

The results showed that GV and VI of nonprimed broccoli seeds at 25˚C were about 85.3% and 27.1, respectively, and the MGT of the nonprimed broccoli seeds was about 1.8 days. At 20, 15, and 10˚C, however, the averaged GV of the nonprimed broccoli seeds reduced to 46.7%, 0%, and 0%, respectively, and the averaged VI reduced to 21.4, 13.7, and 1.1, respectively. In addition, these seeds germinated slower (Fig 1A, 1C, 1E and 1G). The averaged GR of the nonprimed broccoli seeds reduced to 10.7% at 10˚C. In contrast, the SM-primed broccoli seeds showed alleviated reductions of seed vigor under the lower-temperature conditions compared with the nonprimed control seeds. The GRs of the SM-primed broccoli seeds were similar to those of the nonprimed seeds at 25, 20, and 15˚C, and were maintained at 87.3% at 10˚C.

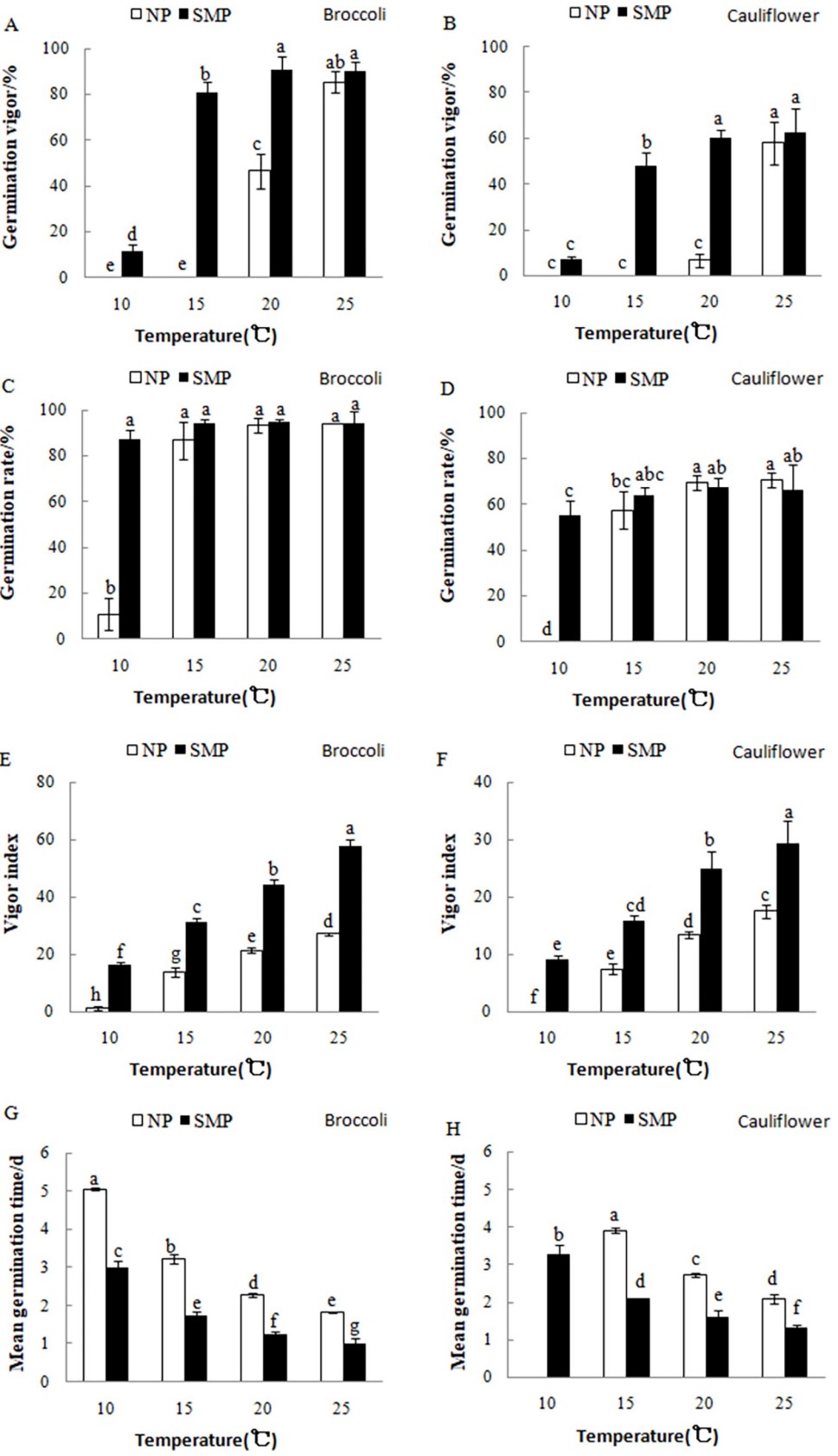

**Fig 1. Effects of SMP on cauliflower and broccoli seed germination at 10, 15, 20, and 25˚C.** (A and B) Germination vigor, (C and D) germination rate, (E and F) vigor index, and (G and H) mean germination time. The nonprimed (NP) and SM-primed (SMP) seeds were germinated on wet filter papers inside plastic boxes for 7 days at 10, 15, 20, and 25˚C, respectively. Different small letters indicate significant differences between the treatments ($P < 0.05$).

The VIs of the SM-primed broccoli seeds were significantly higher than those of the non-primed broccoli seeds, and the MGTs of the SM-primed broccoli seeds were significantly shorter than those of the nonprimed seeds at all assayed temperatures. The GV, GR, and VI of both SM-primed and nonprimed broccoli and cauliflower seeds reduced and the MGT was prolonged with the decrease in germination temperature. The SM-primed and nonprimed cauliflower seeds also showed similar trends. The GV of cauliflower seeds was 0%, 0%, and 6.7% at 10, 15, and 20˚C, respectively. After SMP, the GV increased to 7.3%, 48.0%, and 60.0%, respectively. The GR of SM-primed cauliflower seeds was 55.3% at 10˚C, while the nonprimed seeds did not germinate (Fig 1B, 1D, 1F and 1H).

The root length and fresh weight of both SM-primed and nonprimed broccoli and cauliflower seedlings reduced with the decrease in germination temperature. The root length and fresh weight of both SM-primed and nonprimed broccoli and cauliflower seedlings at 10 and 15˚C significantly reduced compared with those at 20 and 25˚C. The SMP was found to be effective in promoting root length and fresh weight compared with the nonprimed seeds at 10 and 15˚C (Fig 2).

## Effects on antioxidant enzyme activities in germinating seeds

In this study, the SOD activities in the SM-primed germinating broccoli seeds at 25˚C were significantly higher than those in the nonprimed germinating broccoli seeds. At 20, 15, and

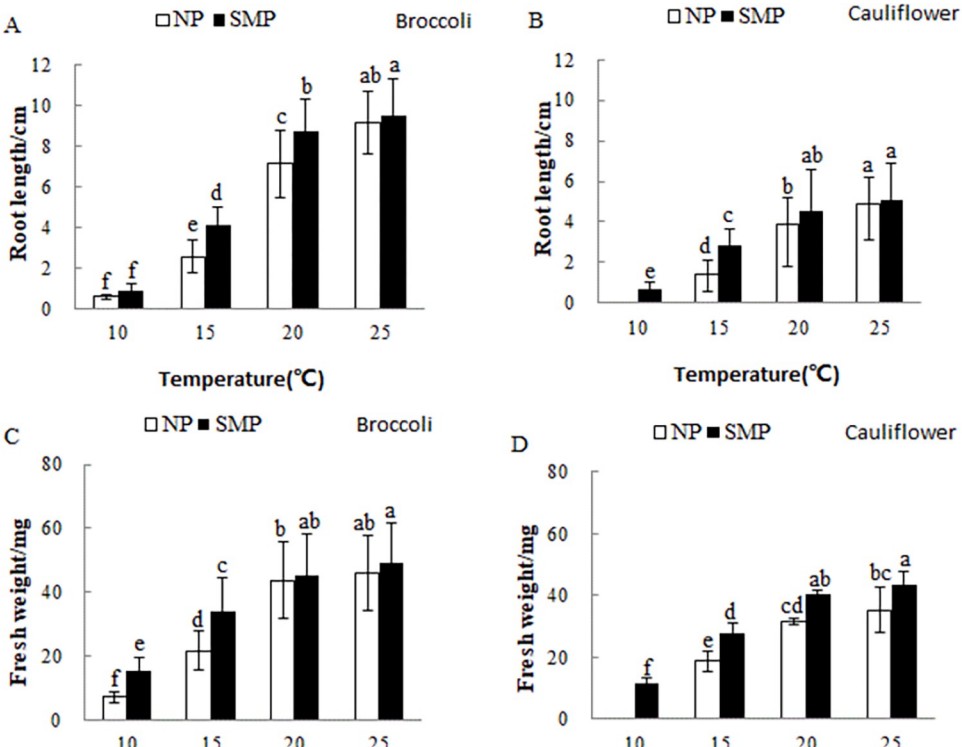

**Fig 2. Effects of SMP on the early growth of cauliflower and broccoli seedlings at 10, 15, 20, and 25˚C.** (A and B) Root length, and (C and D) fresh weight. The seeds of cauliflower and broccoli were germinated as described in Fig 1.

10˚C, the SOD activities in the SM-primed germinating broccoli seeds were similar to those in the nonprimed germinating broccoli seeds. The SOD activities in the SM-primed germinating cauliflower seeds at 25 and 20˚C were similar to those in the nonprimed germinating cauliflower seeds (Fig 3A and 3B). The POD activities in the SM-primed germinating broccoli seeds were consistently higher than those in the nonprimed germinating broccoli seeds, especially at 10, 15, and 20˚C. In the nonprimed germinating cauliflower seeds, the POD activity was always low. The POD activities in the SM-primed germinating cauliflower seeds declined from 25 to 15˚C, and were similar to those in the nonprimed germinating cauliflower seeds at 10˚C (Fig 3C and 3D). The CAT activities in the SM-primed germinating cauliflower and broccoli seeds were always higher than those in the nonprimed germinating cauliflower and broccoli seeds except for SM-primed germinating cauliflower seeds at 10˚C (Fig 3E and 3F). The APX activities in the SM-primed germinating cauliflower and broccoli seeds at 15, 20, and 25˚C were significantly higher than those in the nonprimed germinating cauliflower and broccoli seeds, but were similar to those in the nonprimed germinating cauliflower and broccoli seeds at 10˚C (Fig 3G and 3H).

## Effects on seed emergence

The emergence test using SM-primed and nonprimed broccoli and cauliflower seeds was performed in an early spring greenhouse. In the greenhouse, the daily minimum air temperature ranged from 3.9 to 11.8˚C, and the daily maximum air temperature ranged from 11.4 to 34.7˚C. The SM-primed cauliflower and broccoli seeds emerged earlier and had a higher emergence percentage than the nonprimed cauliflower and broccoli seeds (Table 1 and Fig 4). For example, the emergence percentages of the nonprimed broccoli and cauliflower seeds were 45% and 21%, respectively, while the emergence percentages of the SM-primed broccoli and cauliflower seeds were 77 and 50%, respectively (Table 1). A similar trend was observed for the EI between the SM-primed and nonprimed cauliflower and broccoli seeds. The MET of the SM-primed broccoli and cauliflower seeds significantly reduced compared with that of the nonprimed broccoli and cauliflower seeds.

Seed priming is a widely adopted method to minimize the damages caused by low temperatures during seed germination [10–13, 21–23]. SMP is accomplished by mixing seeds with a solid or semi-solid material and a specified amount of water. During SMP, water is slowly provided to the seeds, and thus slow or controlled imbibition occurs, allowing the repair mechanisms to operate [1]. In the present study, we determined that cauliflower and broccoli seed germination and seedling growth under suboptimal temperatures (10 and 15˚C) could be significantly improved using SMP. Increased oxidative stress is one of the rapid responses under low-temperature conditions. This is associated with increased production of reactive oxygen species (ROS) [23, 24]. Antioxidant enzymes have been shown to scavenge the ROS induced by stresses and various physiological processes during seed development, storage, and germination [25–27]. Our data showed that the POD and APX activities of SM-primed broccoli and cauliflower seeds were generally higher than those of the nonprimed ones when germinated for 17 h for broccoli and 24 h for cauliflower. This finding agreed with previous reports and further confirmed that the priming treatment could improve seed germination performance and increase the activities of multiple antioxidant enzymes. For example, the activities of antioxidant enzymes of primed bitter gourd seeds germinated at suboptimal temperatures increased [10]. SMP treatment also improved the germination of okra seeds at a suboptimal temperature (15˚C) inside the laboratory and reduced the days needed to germinate from 12 to 8.5 days [16]. Hydropriming, osmopriming, and SMP were also reported to enhance the activities of multiple antioxidant enzymes in okra seeds [28]. Primed spinach seeds exhibited a

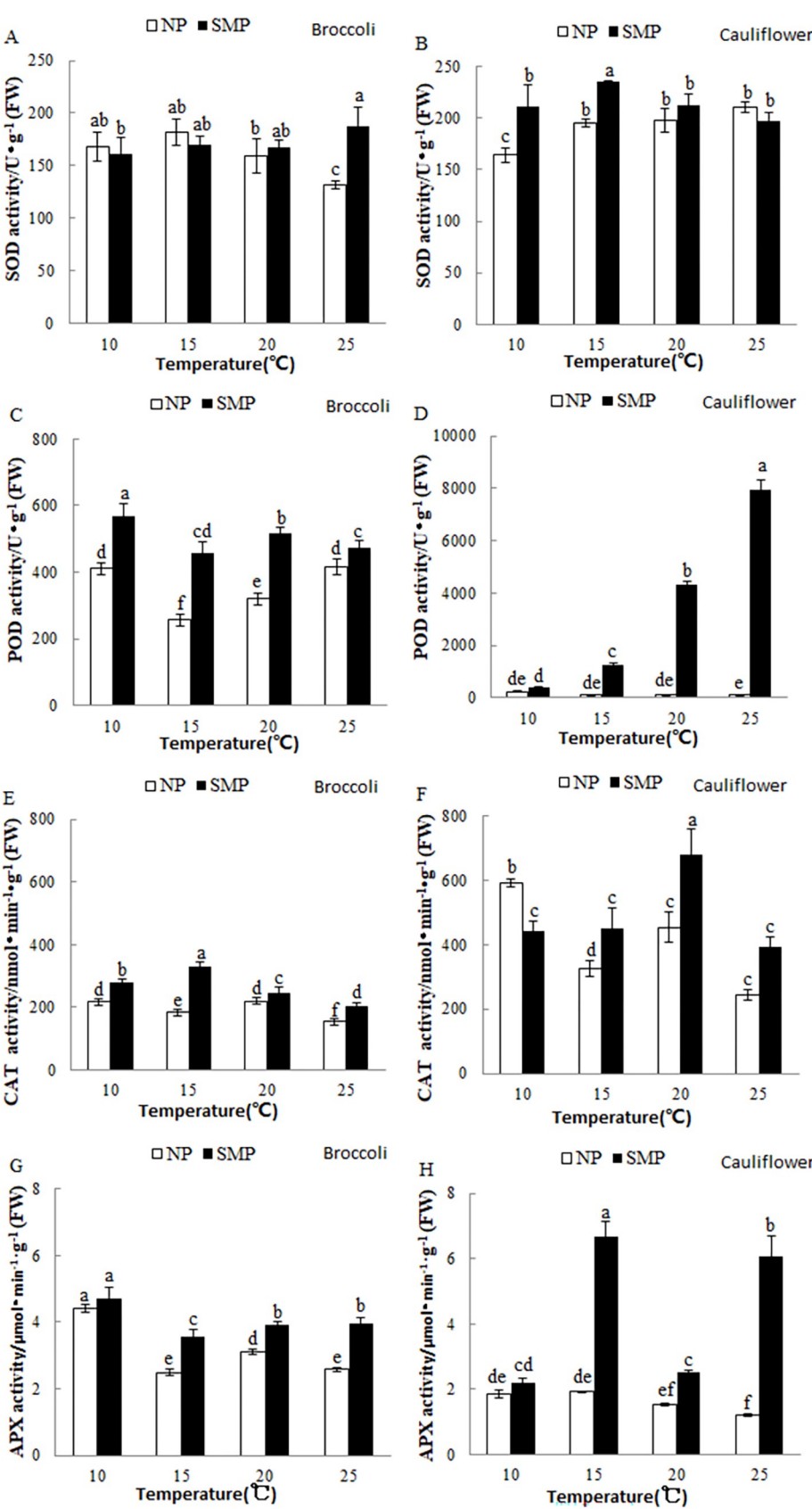

**Fig 3. Effects of SMP on four antioxidant enzymes activities in the germinating cauliflower and broccoli seeds incubated at 10, 15, 20, and 25°C.** (A and B) Superoxide dismutase (SOD) activity, (C and D) peroxidase (POD) activity, (E and F) catalase (CAT) activity, and (G and H) ascorbate peroxidase (APX) activity. In this experiment, cauliflower and broccoli seeds were germinated as described in Fig 1. The activities of SOD, POD, CAT, and APX in the nonprimed (NP) and SM-primed (SMP) germinating seeds were determined 17 h after incubation for broccoli and 24 h after incubation for cauliflower. Each treatment had three biological replicates in this study.

greater increase in APX activity and the accumulation of AsA and GSH, which assisted in early seed germination and seedling establishment, compared with nonprimed control seeds. An early reduction of the CAT activity was observed during germination (0–5 days), followed by a significant increase by 10 days of germination. The primed seeds generally had greater CAT activity than the unprimed controls during the later stage of germination (10–15 days of germination) under normal and chilling conditions. The SOD activity was suppressed during germination [29]. In our study, SM-primed germinating broccoli and cauliflower seeds had greater CAT activity than the nonprimed controls, with only one exception (SM-primed cauliflower seeds at 10°C). The changes in SOD activities in SM-primed broccoli and cauliflower seeds were also somewhat complicated compared with those in nonprimed controls. This might be because both SM-primed and nonprimed germinating broccoli and cauliflower seeds are exiting the protective systems present in dry seeds and CAT and SOD activities might be suppressed in the early germination stage; also, the CAT isoform pattern might vary with physiological conditions [29–31]

Besides seed germination and enzymatic activity assays in the laboratory, we also conducted assays inside the early spring glasshouse. The result showed that the SM-primed cauliflower and broccoli seeds emerged earlier and at a higher emergence percentage compared with the nonprimed cauliflower and broccoli control seeds. The improvement in the emergence of SM-primed seeds might be attributed to the fact that SMP enhanced the POD and APX activities and increased the seed germination potential in the early stage of germination, resulting in increased stress tolerance in germinating seeds. Thus, upon sowing, SM-primed seeds could rapidly imbibe and revive the seed metabolism; also, a renewal of the antioxidant system might be initiated, resulting in a higher GR and faster seedling growth [29, 32]. Consequently, we concluded that the SMP could improve the overall performance of seeds, including faster and more uniform seed germination and better emergence percentage under suboptimal temperatures (10–15°C). Further analyses of the changes in the physiological and molecular pathways in the SM-primed and nonprimed seeds might help reveal the mechanism controlling seed germination and early seedling growth.

## Conclusions

Our results showed that the GV, GR, VI seedling fresh weight, and root length of cauliflower and broccoli seed reduced, especially the seed GR at 10°C, under suboptimal temperature conditions (i.e., 10 and 15°C). It was also observed that the SMP treatment increased GV and VI

**Table 1. Effects of SMP on cauliflower and broccoli seed emergence.**

| Seeds | Treatment | Mean emergence time (day)[#] | Emergence index | Emergence percentage (%) |
|---|---|---|---|---|
| Broccoli | NP | 11.5 | 2.0 | 45.0 |
| | SMP | 7.1* | 5.8* | 77.0* |
| Cauliflower | NP | 13.4 | 0.8 | 21.0 |
| | SMP | 8.3* | 3.4* | 50.0* |

[#], days

*, significant at $P < 0.05$.

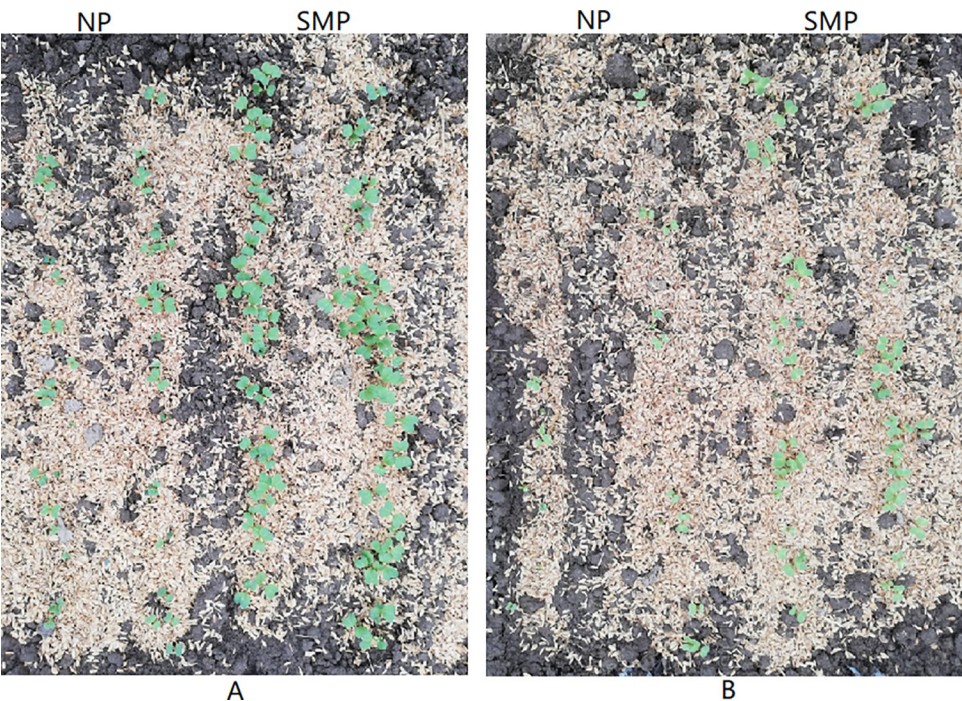

**Fig 4. Images of emergences of nonprimed (NP) or solid matrix primed (SMP) cauliflower and broccoli seeds.** (A) Emergence of broccoli seeds. (B) Emergence of cauliflower seeds. Images were taken 16 days after sowing. The emergence test using nonprimed control seeds and SM-primed seeds of cauliflower and broccoli was performed in a glasshouse. For each treatment, 2 biological replicates with 50 seeds each were sown in a glasshouse.

but decreased MGT compared with that shown by the nonprimed control seeds. In addition, SMP increased the activities of POD and APX in the SM-primed germinating cauliflower and broccoli seeds in the early germination stage. The SM-primed cauliflower and broccoli seeds also showed a better emergence percentage under the suboptimal temperature conditions in the early spring greenhouse. Therefore, we recommended SMP as a useful method to improve seed germination and emergence of cauliflower and broccoli seeds under suboptimal temperature conditions.

## Acknowledgments

The authors would like to thank Dr. X.S. Ding for his help in improving the language of this manuscript.

## Author Contributions

**Data curation:** Yan Jun, Huang Zhi-wu, Wan Yan-Hui.

**Project administration:** Zhu Wei-Min.

**Writing – original draft:** Wu Ling-Yun.

**Writing – review & editing:** Wu Ling-Yun.

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
