## [Decision Letter · Decision Letter 0]

13 Apr 2022

PONE-D-22-05093Solid matrix priming(SMP) improves cauliflower and broccoli seed germination and early growth under the sub-optimal temperature conditionsPLOS ONE

Dear Dr. wu,

Thank you for submitting your manuscript to PLOS ONE. After careful consideration, we feel that it has merit but does not fully meet PLOS ONE’s publication criteria as it currently stands. Therefore, we invite you to submit a revised version of the manuscript that addresses the points raised during the review process.

We look forward to receiving your revised manuscript.

Kind regards,

Umakanta Sarker

Academic Editor

PLOS ONE

Journal Requirements:

"The research was supported by Shanghai green leafy vegetables industry technology system and the Shanghai Agriculture Applied Technology Development Program, China(Grant No.G2016060105). The authors would like to thank Dr. X.S.Ding for his help on the language of this manuscript".

 [The research was supported by Shanghai green leafy vegetables industry technology system and the Shanghai Agriculture Applied Technology Development Program, China(Grant No.G2016060105). The funders had no role in study design,data collection and analysis, decision to publish, or preparation of the manuscript.]

Reviewers' comments:

Reviewer's Responses to Questions

**Comments to the Author**

1. Is the manuscript technically sound, and do the data support the conclusions?

Reviewer #1: Yes

Reviewer #2: Partly

2. Has the statistical analysis been performed appropriately and rigorously? 

Reviewer #1: Yes

Reviewer #2: No

3. Have the authors made all data underlying the findings in their manuscript fully available?

Reviewer #1: Yes

Reviewer #2: Yes

4. Is the manuscript presented in an intelligible fashion and written in standard English?

Reviewer #1: Yes

Reviewer #2: No

5. Review Comments to the Author

Reviewer #1: Manuscript provides useful information in understanding the physiological basis of seed quality enhancement through solid matrix priming under sub-optimal temperatures in cauliflower and broccoli. Authors need to clarify the temperature range for sub-optimal conditions for these crops, since ISTA standards suggest 20 C constant temperature for conducting germination test in cauliflower and broccoli.

I think it would be more appropriate to combine results and discussion for clarity and it will also reduce duplication. The results are too brief and need to be elaborated so that the readers understand the advantages induced by solid matrix priming in improving field performance of cauliflower and broccoli seeds under sub-optimal temperatures.

See manuscript for several other comments, edits and suggestions.

The manuscript is acceptable for publication after thorough revision

Reviewer #2: This is a good peace of research work. Bur methodology and design of experiment should be clear. Discussion has not written properly. Citations are not sufficient language and grammar should be checked and improved.

6. PLOS authors have the option to publish the peer review history of their article (what does this mean?). If published, this will include your full peer review and any attached files.

Reviewer #1: No

Reviewer #2: **Yes: **Professor Dr. M. A. Mannan

---

## [Author Response · Author response to Decision Letter 0]

29 Apr 2022

Response to Reviewer #1, 

Thanks for your comments on our paper. We have revised our paper according to your comments.

10, 15 0C is sub-optimal conditions for conducting germination test in cauliflower and broccoli.

We have combined the results and discussion and revised the results part.

1. 10 oC and15 oC is a sub-0ptimal temperature. 20 oC should be optimal temperature for cauliflower and broccoli seed germination. Needs clarification

Yes, 10 oC and15 oC is a sub-0ptimal temperature. Please see the revised manuscript.

2、Should mention the benefit obtained by SMP under suboptimal temperatures over non primed seed

I have revised. Please see the revised manuscript.

3、Needs elaboration and clarity. 

I have revised. Please see the revised manuscript.

4、Improvement with which method of priming ?

I have revised. Please see the revised manuscript.

5、What was the initial moisture content of seed prior to priming ?

I have revised. Please see the revised manuscript.

6、Follow ISTA protocols for germination test. Specify make of incubators/ germinator used for conducting germination test.

Broccoli and cauliflower seed used in my experiment germinated fast. After germinate for 7 days, there were no new seed germinated.

I have revised others. Please see the revised manuscript.

7、Too brief: the results between primed and non-primed seeds may be compared based on percent increase or decrease.

I have revised others. Please see the revised manuscript.

8、Significantly what? explain

Because change of SOD activity is somewhat complexed in our study, I have deleted this sentence. Please see the revised manuscript.

9、Describe the early spring conditions in terms of min./ max. temperatures. Whether emergence test was conducted outside in the field during early spring. If not then we may say under greenhouse conditions.

I have revised others. Please see the revised manuscript.

10、Mention sub-optimal temperatures.

I have revised. Please see the revised manuscript.

Response to Dr.Mannan, 

Thanks for your comments on our paper. We have revised our paper according to your comments.

1. Mention the design of experiment with number of replication: 

I have revised. Please see the revised manuscript. About number of replication, seed germination test, analysis of antioxidant enzyeme and emergence test have different replications, please see them in materials and methods part. 

2. mention the design of experiment:

 I have revised. Please see the revised manuscript.

3. Detailed results are not presented:

I have revised. Please see the revised manuscript. 

4.Discussion is very poor:

I have revised. Please see the revised manuscript. 

5.Decribe how anti-oxidant activities increased due pring using vermiculite?

I have revised. Please see the revised manuscript. 

5.what are the mechanisms?

I have revised. Please see the revised manuscript. 

6.More citation need

I have revised. Please see the revised manuscript.

---

## [Decision Letter · Decision Letter 1]

20 May 2022

PONE-D-22-05093R1Solid matrix priming(SMP) improves cauliflower and broccoli seed germination and early growth under the sub-optimal temperature conditionsPLOS ONE

Dear Dr. wu,

Thank you for submitting your manuscript to PLOS ONE. After careful consideration, we feel that it has merit but does not fully meet PLOS ONE’s publication criteria as it currently stands. Therefore, we invite you to submit a revised version of the manuscript that addresses the points raised during the review process.

We look forward to receiving your revised manuscript.

Kind regards,

Umakanta Sarker

Academic Editor

PLOS ONE

Journal Requirements:

Reviewers' comments:

Reviewer's Responses to Questions

**Comments to the Author**

1. If the authors have adequately addressed your comments raised in a previous round of review and you feel that this manuscript is now acceptable for publication, you may indicate that here to bypass the “Comments to the Author” section, enter your conflict of interest statement in the “Confidential to Editor” section, and submit your "Accept" recommendation.

Reviewer #2: All comments have been addressed

2. Is the manuscript technically sound, and do the data support the conclusions?

Reviewer #2: Yes

3. Has the statistical analysis been performed appropriately and rigorously? 

Reviewer #2: Yes

4. Have the authors made all data underlying the findings in their manuscript fully available?

Reviewer #2: Yes

5. Is the manuscript presented in an intelligible fashion and written in standard English?

Reviewer #2: Yes

6. Review Comments to the Author

Reviewer #2: (No Response)

7. PLOS authors have the option to publish the peer review history of their article (what does this mean?). If published, this will include your full peer review and any attached files.

Reviewer #2: **Yes: **Professor Dr. M. A. Mannan

---

## [Author Response · Author response to Decision Letter 1]

24 May 2022

Dear Editors:

Thank you very much for your letter and the comments about our paper submitted to PLOS ONE(PONE-D-22-05093R1).

We have checked the manuscript and revised it according to the comments. We submit here the revised manuscript labeled “manuscript” and "revised manuscript with track changes".

Sincerely yours,

Thank you and best regards.

Yours sincerely,

Lingyun Wu 

E-mail:wulingyun@saas.sh.cn

---

## [Editor Report · Decision Letter 2]

31 May 2022

PONE-D-22-05093R2Solid matrix priming(SMP) improves cauliflower and broccoli seed germination and early growth under the sub-optimal temperature conditionsPLOS ONE

Dear Dr. wu,

Thank you for submitting your manuscript to PLOS ONE. After careful consideration, we feel that it has merit but does not fully meet PLOS ONE’s publication criteria as it currently stands. Therefore, we invite you to submit a revised version of the manuscript that addresses the points raised during the review process.

We look forward to receiving your revised manuscript.

Kind regards,

Umakanta Sarker

Academic Editor

PLOS ONE

Additional Editor Comments (if provided):

There are thousands of typos in the MS. Check the grammar of the MS by an English expert.
---

## [Author Response · Author response to Decision Letter 2]

23 Jun 2022

Response to Editor, 

Thanks for your comments on our paper. We have checked and revised the grammar of the manuscript according to your comments. Please see the revised manuscript.

Yours sincerely,

Lingyun Wu 

E-mail:wulingyun@saas.sh.cn

---

## [Editor Report · Decision Letter 3]

24 Jun 2022

PONE-D-22-05093R3Solid matrix priming(SMP) improves cauliflower and broccoli seed germination and early growth under the sub-optimal temperature conditionsPLOS ONE

Dear Dr. wu,

Thank you for submitting your manuscript to PLOS ONE. After careful consideration, we feel that it has merit but does not fully meet PLOS ONE’s publication criteria as it currently stands. Therefore, we invite you to submit a revised version of the manuscript that addresses the points raised during the review process. There are a lots of typos retaining in the MS. I suggest the authors to check the language of the MS by an English Editing service.

We look forward to receiving your revised manuscript.

Kind regards,

Umakanta Sarker

Academic Editor

PLOS ONE
---

## [Author Response · Author response to Decision Letter 3]

21 Jul 2022

Response to reviewers, 

Thanks for your comments on our paper. We have contacted English editing sevice company to check the language according to your comments. 

Please see the revised manuscript.

Yours sincerely,

Lingyun Wu 

E-mail:wulingyun@saas.sh.cn

---

## [Decision Letter · Decision Letter 4]

12 Sep 2022

Solid matrix priming improves cauliflower and broccoli seed germination and early growth under the suboptimal temperature conditions

PONE-D-22-05093R4

Dear Dr. wu,

We’re pleased to inform you that your manuscript has been judged scientifically suitable for publication and will be formally accepted for publication once it meets all outstanding technical requirements.

Kind regards,

Thomas Roach

Academic Editor

PLOS ONE

Additional Editor Comments (optional):

Reviewers' comments:

Reviewer's Responses to Questions

**Comments to the Author**

1. If the authors have adequately addressed your comments raised in a previous round of review and you feel that this manuscript is now acceptable for publication, you may indicate that here to bypass the “Comments to the Author” section, enter your conflict of interest statement in the “Confidential to Editor” section, and submit your "Accept" recommendation.

Reviewer #1: All comments have been addressed

2. Is the manuscript technically sound, and do the data support the conclusions?

Reviewer #1: Yes

3. Has the statistical analysis been performed appropriately and rigorously? 

Reviewer #1: Yes

4. Have the authors made all data underlying the findings in their manuscript fully available?

Reviewer #1: Yes

5. Is the manuscript presented in an intelligible fashion and written in standard English?

Reviewer #1: Yes

6. Review Comments to the Author

Reviewer #1: I think there is an improvement in the language of the manuscript.

This article provides insight of physiological basis of seed quality enhancement through solid matrix priming under sub-optimal temperatures.

Since this article provides useful information, it may therefore be accepted for publication.

7. PLOS authors have the option to publish the peer review history of their article (what does this mean?). If published, this will include your full peer review and any attached files.

Reviewer #1: **Yes: **Vinod Kumar Pandita

---

## [Editor Report · Acceptance letter]

16 Sep 2022

PONE-D-22-05093R4 

Solid matrix priming improves cauliflower and broccoli seed germination and early growth under the suboptimal temperature conditions 

Dear Dr. Ling-Yun:

I'm pleased to inform you that your manuscript has been deemed suitable for publication in PLOS ONE. Congratulations! Your manuscript is now with our production department. 

Kind regards, 

on behalf of

Dr. Thomas Roach 

Academic Editor

PLOS ONE